# Data-Efficient GAN Training Beyond (Just) Augmentations: A Lottery Ticket Perspective

**Tianlong Chen[1], Yu Cheng[2], Zhe Gan[2], Jingjing Liu[3], Zhangyang Wang[1]**
[1]University of Texas at Austin, [2]Microsoft Corporation, [3]Tsinghua University
{tianlong.chen,atlaswang}@utexas.edu,{yu.cheng,zhe.gan}@microsoft.com
JJLiu@air.tsinghua.edu.cn

## Abstract

Training generative adversarial networks (GANs) with limited real image data generally results in deteriorated performance and collapsed models. To conquer this challenge, we are inspired by the latest observation, that one can discover independently trainable and highly sparse subnetworks (a.k.a., lottery tickets) from GANs. Treating this as an inductive prior, we suggest a brand-new angle towards data-efficient GAN training: by first identifying the lottery ticket from the original GAN using the small training set of real images; and then focusing on training that sparse subnetwork by re-using the same set. We find our coordinated framework to offer orthogonal gains to existing real image data augmentation methods, and we additionally present a new feature-level augmentation that can be applied together with them. Comprehensive experiments endorse the effectiveness of our proposed framework, across various GAN architectures (SNGAN, BigGAN, and StyleGAN-V2) and diverse datasets (CIFAR-10, CIFAR-100, Tiny-ImageNet, ImageNet, and multiple few-shot generation datasets). Codes are available at: https://github.com/VITA-Group/Ultra-Data-Efficient-GAN-Training.

## 1 Introduction

The quantity, diversity, and high quality of natural images available in the general domain have played an essential role in the achieved breakthroughs of Generative Adversarial Networks (GANs) [2–11] over the past few years. However, it could become challenging or even infeasible for specific application domains to collect a sufficiently large-scale dataset, due to various constraints on the imaging expense, subject type, image quality, privacy, copyright status, and more. That prohibits GANs' broader applications in these domains, e.g., for generating synthetic training data [12]. Examples of such domains include medical images, images from scientific experiments, images of rare species, or photos of a specific person or landmark. Eliminating the need of immense datasets for GAN training is highly demanded for those scenarios. Naively training GAN with scarce samples leads to overconfident discriminators that overfit the small training data [13–15, 1]; it usually ends up with training divergence and drastic performance degradation (evidenced later in Figure 2).

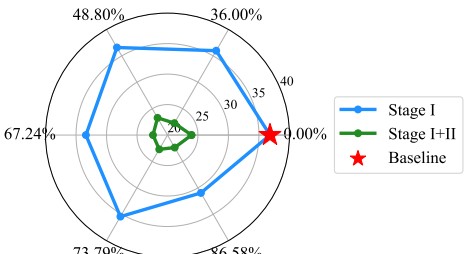

Figure 1: FIDs on training BigGAN on 10% training data from CIFAR-100. Smaller distance to the origin indicates smaller FID/better performance. Compared to the vanilla training baseline (★, i.e., dense model or 0% sparsity), our method's Stage I (●) finds highly sparse lottery tickets from the original BigGAN, with a range of sparsity up to 86.58%. Higher sparsity appears to bring better data-efficiency. Stage II further boosts the training of those found sparse subnetworks, by incorporating existing data-level augmentation [1] and our newly proposed feature-level augmentation (●).

35th Conference on Neural Information Processing Systems (NeurIPS 2021).

This paper addresses the above issue from a brand new perspective by decomposing the challenging GAN training in limited data regimes into two sequential sub-problems: ($i$) finding independent trainable subnetworks (i.e., lottery tickets in GANs) [16, 17]; then ($ii$) training the located subnetworks, which we show is more data-efficient by itself, and can further benefit from aggressive augmentations (both the input data and feature levels). Either sub-problem becomes much less data-hungry to train, and the two sub-problems re-use the same small training set of real images. Although this paper focuses on tackling the data-efficient training of GANs, such a coordinated framework might potentially be generalized to training other deep models with higher data efficiency too.

Our key enabling technique is to leverage the lottery ticket hypothesis (**LTH**) [18]. LTH shows the feasibility to locate highly sparse subnetworks (called "*winning tickets*") that are capable of training in isolation to match or even outperform the performance of original unpruned models. Recently, [17, 16] revealed the existence of winning ticket in GANs (called "*GAN tickets*"). However, none of the existing works discuss the influence of training data size on locating and training those tickets. Our work takes one step further, and shows that one can identify the same high-quality GAN tickets even in the data-scarce regime. The found GAN tickets also serve as a sparse structural prior to solve the second sub-problem with less data, while maintaining an unimpaired *trainability* blessed by the LTH assumption [18]. Figure 1 (the outer circle's blue dots) evidences that we can identify sparse GAN tickets that achieve superior performance than full GANs in the data-scarce scenarios.

The new lottery ticket angle complements the existing augmentation techniques [19–21], and we further show that they can be organically combined to boost performance further[1]. When we train the identified lottery ticket, we demonstrate its training can benefit as well from the latest *data-level* augmentation strategies, ADA [15] and DiffAug [1]. Furthermore, we introduce a novel ***feature-level augmentation*** that can be applied in parallel to data-level. It injects adversarial perturbations into GANs' intermediate features to implicitly regularize both discriminator and generator. Combining the new feature-level and existing data-level augmentations in training GAN tickets leads to more stabilized training dynamics, and establishes new state-of-the-arts for data-efficient GAN training.

Extensive experiments are conducted on a variety of the latest GAN architectures and datasets, which consistently validate the effectiveness of our proposal. For example, our BigGAN tickets at $36.00\%$ and $67.24\%$ sparsity levels reach an (FID, IS) of $(23.14, 52.98)$ and $(70.91, 7.03)$, on Tiny-ImageNet $64 \times 64$ and ImageNet $128 \times 128$, with $10\%$ and $25\%$ training data, respectively. On CIFAR-10 and CIFAR-100, for SNGAN and BigGAN tickets at $67.24\% \sim 86.58\%$ sparsity, our results with only $10\%$ training data can even surpass their dense counterparts. Impressively, our method can generate high-quality images on par with other GAN transfer learning approaches, by training on as few as 100 real samples and without using any pre-training.

## 2 Related Work

**GANs and Data-Efficient GAN Training.** GANs [22] have gained popularity in diverse computer vision scenarios. To stabilize GAN training and improve the visual fidelity and diversity of generated images, extensive studies have been conducted, such as sophisticated network architectures [23, 8, 24–26], improved training recipes [27, 4, 28, 29], and more stable objectives [30–35]. [36, 37] utilize semi- and self-supervised learning to pursue label efficiency in GAN training.

Recently, how to train GANs without sufficient real images in the target domain sparkles new interests. There have been efforts on adapting a pre-trained GAN generator, including BSA [38], AdaFM [39], Elastic Weight Consolidation [40], and Few-Shot GAN [41–43]. However, those methods assume a large, related source domain as *pre-training*, based on which they further alleviate target domain data limitation by only tuning small subsets of weights. They are hence in a **completely different track** from our "stand-alone" data-efficient training goal where no pre-training is leveraged. [44, 45] select core-sets of training data to speed up GAN training. A few recent attempts [1, 15] leverage differentiable or adaptive data augmentations to significantly improve GAN training in limited data regimes. Lately, [46] investigates a regularization approach, on constraining the distance between the current prediction of the real image and a moving average variable that tracks the historical predictions of the generated image, that complements the data augmentation methods.

**Lottery Ticket Hypothesis and GAN Tickets.** [18] claims the existence of independently trainable sparse subnetworks that can match or even surpass the performance of dense networks. [47, 48] scale

---

[1]We also tried to add augmentations in the lottery ticket finding stage, but did not observe visible impact.

up LTH by rewinding [49, 50]. Follow-up researches evidence LTH across broad fields, including visual recognition [18, 47, 51–60], natural language processing [48, 61, 50, 62–64], graph neural network [65], and reinforcement learning [61].

Recently, LTH has been extended to GANs by [16, 17], who validated the existence of winning tickets in the min-max game beyond minimization. Compared with the aforementioned work, our work is the first to study LTH in the data-scarce regime (for GANs, and in general). Besides finding highly compact yet same capable subnetworks, our work reveals LTH's power in saving training data - an appealing perspective never being examined before.

**Adversarial Training and Augmentations.** Deep neural networks suffer from severe performance degradation [66, 67] when facing adversarial inputs [67–69]. To address this notorious vulnerability, various defense mechanisms [70–79] have been proposed. Among others, adversarial training-based approaches achieve superior adversarial robustness [67–69], although at the price of sacrificing benign generalization [80, 70–75].

Several recent works investigate enhancing model (benign) generalization ability with adversarial training [81–86]. They adopt adversarially perturbed input images, embeddings, or intermediate features, into model training to ameliorate performance on the clean test sets. Specifically, the damaging effects of adversarial training could be controlled by extra batch normalization [81] or so. Different from those minimization problems that previous work has focused on, the two-player GAN optimization is more challenging. Generally, the adversarial competition between two players poses impediments to exploit extra adversarial information during training GANs.

## 3 Methodology

### 3.1 Revisiting GANs and the Overfitting Challenge

Generative adversarial networks (GANs) are dedicated to modeling the target distribution with the two-player game formulation of a generator $\mathcal{G}$ and a discriminator $\mathcal{D}$. Specifically, the generator $\mathcal{G}$ takes a random sampled latent vector $\boldsymbol{z}$ (e.g., from a Gaussian distribution) as input and outputs the fake sample $\mathcal{G}(\boldsymbol{z})$. The discriminator $\mathcal{D}$ aims to distinguish generated fake samples $\mathcal{G}(\boldsymbol{z})$ from real samples $\boldsymbol{x}$. Alternative optimizations for the discriminator's loss $\mathcal{L}_{\mathcal{D}}$ and the generator's loss $\mathcal{L}_{\mathcal{G}}$ are adopted in the standard GAN training, which can be depicted as follows:

$$
\begin{aligned}
\mathcal{L}_{\mathcal{D}} &:= \mathbb{E}_{\boldsymbol{x}\sim p_{\mathrm{data}}(\boldsymbol{x})}[f_{\mathcal{D}}(-\mathcal{D}(\boldsymbol{x}))] + \mathbb{E}_{\boldsymbol{z}\sim p(\boldsymbol{z})}[f_{\mathcal{D}}(\mathcal{D}(\mathcal{G}(\boldsymbol{z})))] \\
\mathcal{L}_{\mathcal{G}} &:= \mathbb{E}_{\boldsymbol{z}\sim p(\boldsymbol{z})}[f_{\mathcal{G}}(-\mathcal{D}(\mathcal{G}(\boldsymbol{z})))],
\end{aligned}
$$

where loss functions $f_{\mathcal{D}}(x), f_{\mathcal{G}}(x)$ have multiple choices, e.g., the non-saturating loss [3] with $f_{\mathcal{D}}(x) = f_{\mathcal{G}}(x) = \log(1+e^x)$, and the hinge loss [23] with $f_{\mathcal{D}}(x) = \max(0, 1+x)$ and $f_{\mathcal{G}}(x) = x$. $p_{\mathrm{data}}(\boldsymbol{x})$ and $p(\boldsymbol{z})$ represent the data distribution of real samples and latent vectors. $\mathcal{L}_{\mathcal{D}}$ is maximized to update $\mathcal{D}$'s parameters $\phi$ (i.e., $\mathcal{D}(\cdot) := \mathcal{D}(\cdot, \phi)$), and $\mathcal{L}_{\mathcal{G}}$ is minimized to update $\mathcal{G}$'s parameters $\theta$ (i.e., $\mathcal{G}(\cdot) = \mathcal{G}(\cdot, \theta)$).

---

**Algorithm 1** Data-Efficient Iterative Magnitude Pruning Procedures

---

1: **Input:** Initial two masks $m_g = 1^{\|\theta\|_0}$ and $m_d = 1^{\|\phi\|_0}$; Initialization weights $\theta_0$ and $\phi_0$
2: **Output:** $\{\mathcal{G}(\cdot, \theta_0 \odot m_g), \mathcal{D}(\cdot, \phi_0 \odot m_d)\}$
3: **repeat**
4:   Training $\{\mathcal{G}(\cdot, \theta_0 \odot m_g), \mathcal{D}(\cdot, \phi_0 \odot m_d)\}$ for $t$ epochs *with limited training data*
5:   Pruning $\rho = 20\%$ of remaining weights in both $\mathcal{G}$ and $\mathcal{D}$
6:   Updating the binary masks $m_g$ and $m_d$ accordingly
7:   Rewinding weights of $\mathcal{G}, \mathcal{D}$ to $\theta_0$ and $\phi_0$
8: **until** masks reach the desired sparsity level

---

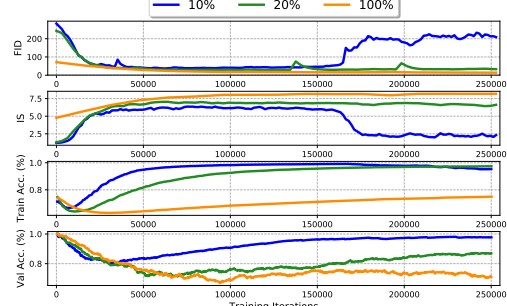

Figure 2: The performance of SNGAN heavily degrades with limited amount training data. Top two figures show that training with 10% of CIFAR-10 data incurs Fréchet Inception Distance (FID) explosion and Inception Score (IS) drop, with the model (**blue curves**) collapsed. Bottom two figures present $\mathcal{D}$'s training and validation accuracies of correctly predicting generated images as fake samples.

**Training Failures of GANs under Limited Data.** [15, 1] observe that GANs' performance has severely deteriorated when only limited training data is available. The discriminator tends to memorize and heavily *overfit* the small training samples, leaving a gap between the real sample's and the generated sample's distribution. As shown in Figure 2, with only $10\%$ CIFAR-10 data available for training SNGAN [23], the training and validation accuracies of the discriminator $\mathcal{D}$ quickly saturate to nearly $100\%$ (ideally close to $50\%$), which indicates $\mathcal{D}$ to become over-confident in distinguishing real and generated samples. It demonstrates that $\mathcal{D}$ simply memorizes the training data, and such overfitting leads to training collapses and deteriorated quality of generated images.

To address this dilemma, we suggest a new data-efficient GAN training workflow, decomposed into two stages: ($i$) *finding winning tickets in GANs* via Algorithm 1; then ($ii$) *training the found GAN tickets*, potentially with both data- and feature-level augmentations, via Algorithm 2. Blessed by LTH, the located GAN ticket shows improved generalization ability, and is further enhanced by augmentations that prevent $\mathcal{D}$ from becoming too confident.

### 3.2 Data-Efficient Lottery Ticket Finding from GANs

In this section, we provide the preliminaries and setups to identifying data-efficient GAN tickets.

**Subnetworks and winning tickets.** A subnetwork of GAN is defined as $\{\mathcal{G}(\cdot, \theta \odot m_g), \mathcal{D}(\cdot, \phi \odot m_d)\}$, where $m_g \in \{0,1\}^{\|\theta\|_0}$ and $m_d \in \{0,1\}^{\|\phi\|_0}$ are binary masks for the generator and discriminator respectively, and $\odot$ is the element-wise product. Let $\theta_0$ and $\phi_0$ be the initialization weights of GANs. Following [18, 16], we define *winning tickets* of GAN as subnetworks $\{\mathcal{G}(\cdot, \theta_0 \odot m_g), \mathcal{D}(\cdot, \phi_0 \odot m_d)\}$, that reach a matched or better performance compared to unpruned GANs when trained in isolation with similar training iterations.

**Finding data-efficient winning tickets in GANs.** To our best knowledge, we are the first to extend LTH to the limited data regimes. In this challenging scenario, *only a small amount of training data are accessible for the finding and training of GAN tickets*. We use unstructured magnitude pruning [87], e.g., Iterative Magnitude Pruning (IMP), to establish the sparse masks $m_g$ and $m_d$.

As shown in Algorithm 1, we first train the full GAN model for $t$ epochs with limited training samples (e.g., 100-shot), and then perform IMP to globally prune the weights with the lowest magnitude. Zero elements in the obtained masks $m_g$ and $m_d$ index the pruned weights. Before repeating the process again, the weights of the sparse generator $\mathcal{G}(\cdot, \theta \odot m_g)$ and discriminator $\mathcal{D}(\cdot, \phi \odot m_d)$ are rewound to the same initialization $\theta_0$ and $\phi_0$, following the convention [18]. The pruning ratio $\rho$ controls the portion of weights removed per round, and we fix $\rho = 20\%$ in all experiments.

Intuitively, identifying a special sparse mask (without requiring to train its weights well) should be an easier and hence more data-efficient task compared to training the full network weights. That was verified by our observations in experiments too: when the training data volume reduces from $100\%$ to $10\%$ of the full training set, the quality of sparse mask remains to be stable, since it achieves matched performance compared to its dense counterpart in both full and limited data re-training regimes.

### 3.3 Data-Level and Feature-level Augmentations for Training GAN Tickets

After locating the GAN ticket at certain sparsity, training it using the vanilla recipe could already attain significantly improved IS and FID compared to the full dense model trained in the same data-limited regime: see Figure 1 outer circle for example.

Next, we discuss how our proposal can be applied together with augmentation-based approaches, for enhanced training of our found GAN tickets. Our natural choices include to plug-in the two recent state-of-the-art data augmentations, i.e., DiffAug [1] and ADA [15]. We further present a new adversarial *feature-level* augmentation (AdvAug), that can be jointly applied together with data-level augmentations to gain an additional performance boost.

**Revisiting adversarial training.** Let $(\boldsymbol{x}, \boldsymbol{y})$ denote the input image and its label. $f(\vartheta, \boldsymbol{x}, \boldsymbol{y})$ is the loss function parameterized by $\vartheta$. Adversarial training [69] can be formulated as follows:

$$\min_{\vartheta} \mathbb{E}_{(\boldsymbol{x}, \boldsymbol{y})} \left[ \max_{\|\delta\|_{\mathrm{p}} \leq \epsilon} f(\vartheta, \boldsymbol{x} + \delta, \boldsymbol{y}) \right], \tag{1}$$

where $\delta$ is crafted adversarial perturbation constrained within the $\ell_{\mathrm{p}}$ norm ball that is centered at $x$ with a radius $\epsilon$. $\delta$ can be reliably generated by multi-step projected gradient descent (PGD) [69].

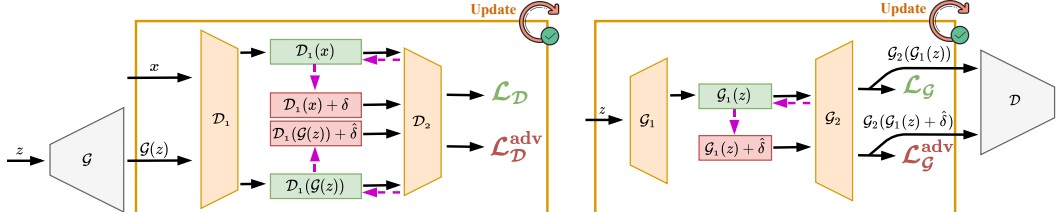

Figure 3: The pipeline of AdvAug for GANs. *Left:* Updating the discriminator $\mathcal{D}$; *Right:* Updating the generator $\mathcal{G}$. Purple arrows denote the path to generate adversarial feature perturbations.

Different from the above standard adversarial training, which adds perturbation on the image pixel space, AdvAug injects adversarial perturbations to intermediate feature embeddings of both $\mathcal{G}$ and $\mathcal{D}$. A similar feature augmentation idea was proven to be helpful in NLP [82], and computer vision [86], showing effectiveness to regularize the smoothness of the training landscape and enhance the trained model's generalization. The AdvAug scheme is illustrated in Figure 3, and can be mathematically depicted as follows ($\lambda_1$ is a controlling hyperparameter):

$$\min_{\theta} \mathcal{L}_{\mathcal{G}} + \lambda_1 \cdot \mathcal{L}_{\mathcal{G}}^{\text{adv}} \ s.t. \ \mathcal{L}_{\mathcal{G}}^{\text{adv}} := \max_{\|\hat{\delta}\|_{\infty} \leq \epsilon} \mathbb{E}_{\boldsymbol{z} \sim p(\boldsymbol{z})}[f_{\mathcal{G}}(-\mathcal{D}(\mathcal{G}_2(\mathcal{G}_1(\boldsymbol{z}) + \hat{\delta})))]. \tag{2}$$

We choose $\lambda_1 = 1$ in all experiments for simplicity. $\mathcal{G} = \mathcal{G}_2 \circ \mathcal{G}_1$ denotes the generator, and between the $\mathcal{G}_1$ and $\mathcal{G}_2$ parts we inject AdvAug. Adversarial perturbations $\hat{\delta}$ generated by PGD [69], are applied to the intermediate feature space $\mathcal{G}_1(\boldsymbol{z})$. The details of our feature-level augmentation on the discriminator $\mathcal{D}$ is included in Appendix A1. The full algorithm of training GAN with both **data-and feature-level augmentations** is summarized in Algorithm 2.

---

**Algorithm 2** Training (Sparse) GAN with Data- and Feature-level Augmentations

---

  **Input:** GAN $\{\mathcal{G}(\cdot, \theta_0), \mathcal{D}(\cdot, \phi_0)\}$; Inputs $\boldsymbol{x}$ and $\boldsymbol{z}$
  **Output:** Trained GAN $\{\mathcal{G}(\cdot, \theta_{\text{T}}), \mathcal{D}(\cdot, \phi_{\text{T}})\}$
  **for** $t = 1$ **to** T **do**
    # Training discriminator with data and feature augmentations
    Augment input with DiffAug [1] or ADA [15]
    Feed $\boldsymbol{x}$ and $\mathcal{G}(\boldsymbol{z})$ to $\mathcal{D}$
    Generate adversarial augmented features in $\mathcal{D}$ ( Equation. 5)
    Update the discriminator $\mathcal{D}(\cdot, \phi_t)$ (Equation. 6)
    # Training generator with data and feature augmentations
    Sample and augment $\boldsymbol{z}$ with DiffAug [1] or ADA [15]
    Feed $\boldsymbol{z}$ to $\mathcal{G}$. Generate adversarial augmented features in $\mathcal{G}$ ( Equation. 3)
    Update the discriminator $\mathcal{G}(\cdot, \theta_t)$ (Equation. 4)
  **end for**

---

Note that AdvAug only affects the generated images through $\mathcal{G}$ intermediate features, and the classifier learning through $\mathcal{D}$ features. It hence avoids to directly manipulate the real data distribution. One bonus of doing so is that it is potentially better at alleviating the distribution leaking issue [15], i.e., GANs learn to mimic and generate the augmented distribution rather than the real one.

## 4 Experiments

In this section, we conduct comprehensive experiments on Tiny-ImageNet [88], ImageNet [89], CIFAR-10 [90], and CIFAR-100 based on the unconditional SNGAN [23] and StyleGAN-V2 [6], as well as the class-conditional BigGAN [2]. We adopt the common evaluation metrics, including Fréchet Inception Distance (FID) [91] and Inception Score (IS) [34]. Note that a smaller FID ($\downarrow$) and a larger IS ($\uparrow$) indicate better performing GAN models. Furthermore, we evaluate our proposed method on few-shot generation both with and without pre-training in Section 4.3. Extensive ablation studies analyze effectiveness of each component in Section 4.4.

**Implementation and Baseline Details.** We follow the popular StudioGAN codebase [92], which contains high-quality re-implementation of BigGAN and SNGAN on ImageNet and CIFAR. For example, our implemented BigGAN baseline performs much better, i.e., FID: 26.44 (ours) v.s. 39.78 (reported) on CIFAR-10, and FID: 36.58 (ours) v.s. 66.71 (reported) on CIFAR-100, than the recent reported baselines in [1], under $10\%$ training data regimes. For detailed configuration, BigGAN takes

learning rates of $\{4, 2, 2\} \times 10^{-4}$ for $\mathcal{G}$, of $\{1, 5, 2\} \times 10^{-4}$ for $\mathcal{D}$, batch sizes of $\{256, 256, 64\}$, $1 \times 10^5$ training iterations, and $\{1, 2, 5\}$ $\mathcal{D}$ steps per $\mathcal{G}$ step on {Tiny-ImageNet, ImageNet, CIFAR} datasets. SNGAN uses learning rates of $2 \times 10^{-4}$ for $\mathcal{G}$ and $\mathcal{D}$, batch sizes of 64, $5 \times 10^4$ training iterations, and five $\mathcal{D}$ steps per $\mathcal{G}$ step on CIFAR. For StyleGAN-V2 experiments, we use its popular PyTorch implementation[2], and keep the default configuration in [1] including image resolution ($256 \times 256$), learning rates for $\mathcal{D}/\mathcal{G}$ ($2 \times 10^{-4}$), batch size (5), and training iterations ($1 \times 10^5$).

Note that, same as the setting in [1], training iterations will be doubled when training GANs with DiffAug. We use implementations in the StudioGAN codebase for DiffAug [1], and the official implementation[3] for ADA [15]. AdvAug with PGD-1 and step size $0.01/0.001$ is applied on CIFAR/(Tiny-)ImageNet datasets, which are tuned by a grid search in Section 4.4. All GANs are trained with 8 pieces of NVIDIA V100 32GB.

## 4.1 On the Effectiveness of Training with Winning Ticket and AdvAug

Table 1: **Tiny-ImageNet** $64 \times 64$ performance without the truncation trick [2]. FID and IS are measured using 10K samples; the official validation set is utilized as the reference distribution. BigGANs at $0.00\%$ (full unpruned models), $36.00\%, 67.24\%$ sparsity are found and trained with $100\%, 20\%, 10\%$ data, respectively.

| Methods | 100% training data (full set) | | 20% training data | | 10% training data | |
|---|---|---|---|---|---|---|
| | FID ($\downarrow$) | IS ($\uparrow$) | FID ($\downarrow$) | IS ($\uparrow$) | FID ($\downarrow$) | IS ($\uparrow$) |
| BigGAN (0.00%) | $21.54 \pm 0.03$ | $18.33 \pm 0.15$ | $59.77 \pm 0.05$ | $7.81 \pm 0.20$ | $84.53 \pm 0.08$ | $5.45 \pm 0.23$ |
| + AdvAug | $21.07 \pm 0.03$ | $18.92 \pm 0.09$ | $58.55 \pm 0.05$ | $8.46 \pm 0.19$ | $81.72 \pm 0.05$ | $6.32 \pm 0.18$ |
| BigGAN (36.00%) | $20.54 \pm 0.05$ | $18.42 \pm 0.20$ | $59.56 \pm 0.04$ | $7.98 \pm 0.20$ | $75.76 \pm 0.08$ | $6.49 \pm 0.21$ |
| + AdvAug | $\mathbf{20.02 \pm 0.04}$ | $\mathbf{19.15 \pm 0.18}$ | $58.24 \pm 0.05$ | $8.55 \pm 0.20$ | $71.47 \pm 0.07$ | $6.86 \pm 0.20$ |
| BigGAN (67.24%) | $26.37 \pm 0.03$ | $16.38 \pm 0.15$ | $59.02 \pm 0.03$ | $8.17 \pm 0.18$ | $73.23 \pm 0.05$ | $6.68 \pm 0.15$ |
| + AdvAug | $25.59 \pm 0.03$ | $17.62 \pm 0.16$ | $\mathbf{57.60 \pm 0.04}$ | $\mathbf{8.94 \pm 0.18}$ | $\mathbf{70.91 \pm 0.05}$ | $\mathbf{7.03 \pm 0.16}$ |

We adopt the top-performing model BigGAN [2], and report experiments on both Tiny-ImageNet at $64 \times 64$ resolution and ImageNet at $128 \times 128$ resolution. We evaluate our proposal on Tiny-ImageNet with $10\%, 20\%, 100\%$ data available, and ImageNet with $25\%$ data available, with results summarized in Tables 1 and 2, respectively. All our results are averaged over three independent evaluation runs (same hereinafter), and the best performance of each column are highlighted.

The following observations can be drawn: *First*, sparse BigGAN tickets can achieve consistently improved performance over the full model ($0.00\%$). Especially, with only $10\%$ training data available, BigGAN tickets at $67.24\%$ sparsity obtain massive gains of 11.30 FID and 1.58 IS on Tiny-ImageNet. *Second*, feature-level augmentation (AdvAug) consistently improves all training cases, from dense to sparse, and from full data to limited data. In particular, larger sparsity (e.g., $67.24\%$) with less training data available (e.g., $10\%$) tend to benefit more from applying AdvAug, which is aligned with our design principle. *Third*, while the full data regime ($100\%$ data) does not necessarily prefer the highest sparsity (moderate sparsity still benefits), the limited data regimes ($10\%$ or $20\%$ data) see monotonically increasing gains as the ticket sparsity goes higher. That is understandable since the former may need more model capacity to absorb full training data, while the latter case hinges on sparsity to avoid overfitting their limited training data.

Besides, we report another group of experiments of training SNGAN on CIFAR-10, using from $10\%$ to $90\%$ training data. The results are summarized in Figure 4. The conclusions we can draw are highly consistent with the above BigGAN case: (1) at the same data availability (from $10\%$ to even $90\%$), training a sparse ticket is always preferred over training the dense model; (2) AdvAug is also consistently helpful in all cases; (3) for both sparsity and AdvAug, they can contribute to larger gains when training data gets smaller.

Table 2: **ImageNet** $128 \times 128$ performance without the truncation trick [2]. FID and IS are measured using 50K samples; the validation set is utilized as the reference distribution. BigGANs at $0.00\%$ and $36.00\%$ sparsity levels are adopted, and only $25\%$ training data are available in all training stages.

| Methods | 25% training data | |
|---|---|---|
| | FID ($\downarrow$) | IS ($\uparrow$) |
| BigGAN (0.00%) | $25.37 \pm 0.07$ | $46.50 \pm 0.40$ |
| + AdvAug | $23.95 \pm 0.06$ | $47.95 \pm 0.32$ |
| BigGAN (36.00%) | $24.03 \pm 0.08$ | $50.07 \pm 0.51$ |
| + AdvAug | $\mathbf{23.14 \pm 0.07}$ | $\mathbf{52.98 \pm 0.47}$ |

[2]https://github.com/lucidrains/StyleGAN-V2-pytorch. Note that the PyTorch version remains with a small performance gap compared to the TensorFlow implementation in [1].

[3]https://github.com/NVlabs/StyleGAN-V2-ada-pytorch. The official Pytorch implementation in [15].

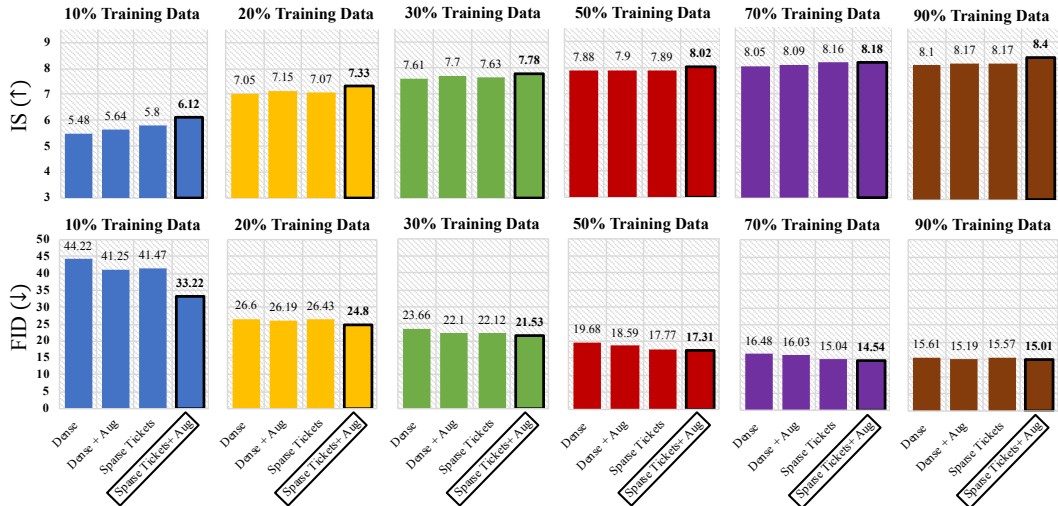

Figure 4: IS (↑) and FID (↓) results of SNGAN with $10\%, 20\%, 30\%, 50\%, 70\%, 90\%$ training data of **CIFAR-10**. Four settings are evaluated:(*i*) Dense (unpruned SNGAN), (*ii*) Dense+Aug (we only apply AdvAug here), (*iii*) Sparse Tickets (pruned SNGAN), (*iv*) Sparse Tickets+Aug, where the top performing variants are highlighted with **black boxes**. SNGAN tickets with $20\%, 36\%, 36\%, 67\%, 49\%, 36\%$ sparsity levels are adopted accordingly. IS and FID are measured using 10K samples; the validation set is utilized as the reference.

Table 3: **CIFAR-10 and CIFAR-100** results. FID (↓) are measured using 10K samples; the validation set is utilized as the reference distribution. Full dense models and sparse winning tickets of BigGAN are reported with $100\%, 20\%, 10\%$, respectively. Specifically, BigGAN tickets with $67.24\%$ and $86.58\%$ sparsity levels are reported for the $100\%, 20\%$ and $10\%$ training data regimes. Performance reported is averaged over three independent evaluation runs; all standard deviation are less than $1\%$.

| Methods | CIFAR-10 | | | CIFAR-100 | | |
|---|---|---|---|---|---|---|
| | 100% data | 20% data | 10% data | 100% data | 20% data | 10% data |
| **Dense** BigGAN | 8.57 | 17.38 | 26.44 | 11.83 | 22.13 | 36.58 |
| + DiffAug [1] | 8.09 | 13.04 | 17.40 | 10.60 | 18.32 | 25.69 |
| + DiffAug + AdvAug | **7.70** | 12.19 | 14.40 | **8.96** | 17.94 | 23.94 |
| **Sparse** BigGAN Tickets | 8.26 | 16.03 | 25.41 | 11.73 | 21.05 | 30.96 |
| + DiffAug [1] | 8.19 | 12.83 | 16.74 | 10.73 | 17.43 | 23.80 |
| + DiffAug + AdvAug | 8.15 | **12.02** | **14.38** | 10.14 | **17.19** | **22.37** |

## 4.2 Incorporating Our Proposal with Latest Data Augmentations and Regularization

**Combining DiffAug.** We first incorporate DiffAug [1], as a representative of the latest data-level augmentation, into our proposal and show the complementary gains. We conduct experiments on the class-conditional BigGAN and unconditional SNGAN models with CIFAR-10 and CIFAR-100. For BigGAN, we utilize $100\%, 20\%, 10\%$ data to locate GAN tickets; then we train them with data-level DiffAug, or with both DiffAug and feature-level AdvAug, as shown in Table 3. Consistent observations can be drawn: *First*, similar to our previous observations on AdvAug, DiffAug also shows to contribute more when the training data becomes more limited; *Second*, combining DiffAug and AdvAug improves over either alone, and leads to the best results across all cases.

Table 4: **CIFAR-100** results with only $10\%$ training data available. FID (↓) of three evaluation runs are measured using 10K samples; the validation set is utilized as the reference distribution. Sparse StyleGAN-V2 tickets at $48.80\%$ sparsity are adopted.

| Methods | 10% training data |
|---|---|
| **Dense** StyleGAN-V2 | $13.59 \pm 0.06$ |
| + DiffAug [1] | $12.90 \pm 0.04$ |
| + ADA [15] | $12.87 \pm 0.03$ |
| + ADA + $R_{LC}$ [46] | $13.01 \pm 0.02$ |
| **Sparse** StyleGAN-V2 Tickets | $13.05 \pm 0.07$ |
| + DiffAug [1] | $12.53 \pm 0.03$ |
| + ADA [15] | $12.20 \pm 0.03$ |
| + ADA + $R_{LC}$ [46] | $12.48 \pm 0.04$ |
| + ADA [15] + AdvAug | $\mathbf{12.11 \pm 0.05}$ |

**Combining Other Data Augmentations and Regularization.** We then extend our combination study to other recent data augmentation and regularization approaches, e.g., ADA [15] and $R_{LC}$ [46].

Experiments are conducted on CIFAR-100 with StyleGAN-V2 backbone, and results are collected in Table 4. We observe that plugging in either ADA [15] or DiffAug [1] into our framework could improve sparse GAN winning tickets, and the gain is also enlarged when ADA is combined with AdvAug. Regard to $R_{\mathrm{LC}}$[4], it is less effective combined with other augmentations.

Taking above together, it has been clearly shown that our proposal is orthogonal to those existing efforts and is of independent merit. Moreover, combing them would lead to more powerful pipelines for data-efficient GAN training.

## 4.3 Few-Shot Generation

It is laborious, and sometimes impossible to collect a large-scale dataset for certain images of interest. To tackle the few-shot image generation problem, [93] utilizes pre-training from external large-scale datasets and performs fine-tuning under limited data scenarios; [94], [38] and [95] partially fine-tune the GANs with part of the GAN model being frozen.

We compare these transfer learning approaches[5] with our data-efficient training scheme. **Differently from them**, ours is training from scratch and is free of any pre-training, while all transfer learning methods start from a pre-trained StyleGAN-V2 model on the FFHQ face dataset [5].

Our comparison experiments are conducted using StyleGAN-V2 on the AnimalFace [96] dataset (160 cats and 389 dogs), and the 100-shot Obama, Grumpy Cat, and Panda datasets provided by [1]. As shown in Table 5, our method finds data-efficient GAN tickets at $48.80\%$ sparsity levels, that can be trained with only 100 training samples from scratch (*without any pre-training*) and show competitive performance to other transfer learning algorithms. Visualizations of style space interpolation and few-shot generation are provided in Figure 5 and 6.

Table 5: **Few-shot generation.** Following the setting in [1], we calculate the FID with 5K samples and the training dataset is adopted as the reference distribution. All transfer learning methods have their pre-trainings from FFHQ [5]. StyleGAN-V2 tickets at $48.80\%$ sparsity level are found and used in our method.

| Methods | Pre-training? | 100-shot by [1] | | | AnimalFace | |
| --- | --- | --- | --- | --- | --- | --- |
| | | Obama | Grumpy Cat | Panda | Cat | Dog |
| Scale/shift [38] | Yes | 50.72 | 34.20 | 21.38 | 54.83 | 83.04 |
| MineGAN [95] | Yes | 50.63 | 35.54 | 14.84 | 54.45 | 93.03 |
| TransferGAN [93] | Yes | 48.73 | 34.06 | 23.20 | 52.61 | 82.38 |
| FreezeD [94] | Yes | 41.87 | 31.22 | 17.95 | 47.70 | 70.46 |
| StyleGAN-V2 (0.00%) | No | 89.18 | 61.97 | 90.96 | 95.75 | 164.54 |
| + DiffAug + AdvAug | No | 54.11 | 35.46 | 15.94 | 54.02 | 72.47 |
| StyleGAN-V2 Tickets (48.80%) | No | 73.92 | 56.81 | 82.45 | 85.92 | 153.90 |
| + DiffAug + AdvAug | No | **52.86** | **31.02** | **14.75** | **47.40** | **68.28** |

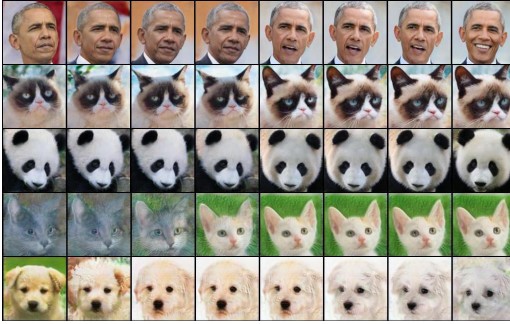
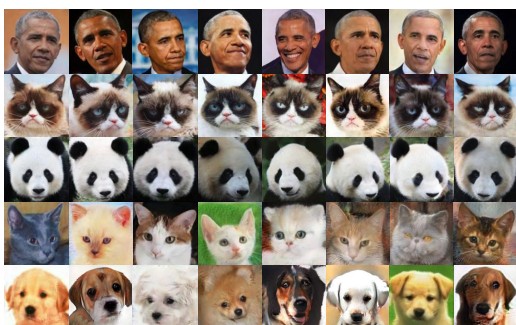

Figure 5: Style interpolation visualizations of StyleGAN-V2 tickets (48.80%) with AdvAug only on 100-shot Obama, Grumpy Cat, Panda, and AnimalFace datasets, respectively.

Figure 6: Few-shot generalization results of StyleGAN-V2 tickets (48.80%) with AdvAug only on 100-shot Obama, Grumpy Cat, Panda, and AnimalFace datasets, respectively.

---

[4][46] advocates the best-performing configuration is ADA+$R_{\mathrm{LC}}$, with FID 13.01 on 10% data of CIFAR-100.
[5]Implementations are from the codebase of [94].

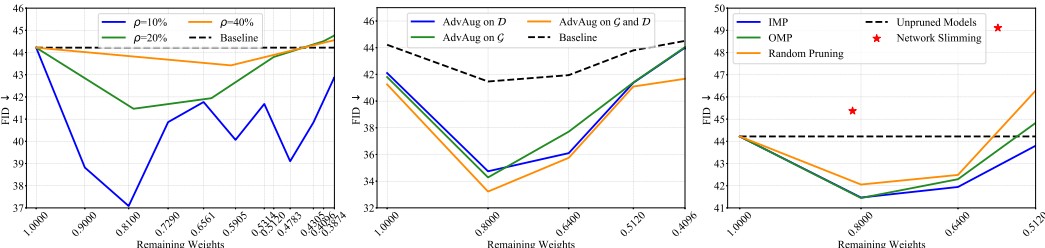

Figure 7: Performance of training GAN with 10% data of CIFAR-10. Remaining weight indicates the sparsity levels of identified GAN tickets. *Left*: FID of the data-efficient GAN tickets found by IMP with pruning ratios $\rho = 10\%, 20\%, 40\%$. *Middle*: FID of the data-efficient GAN tickets trained with different settings of AdvAug, including baseline without AdvAug, AdvAug on $\mathcal{D}$ or $\mathcal{G}$ only, and AdvAug on both $\mathcal{G}$ and $\mathcal{D}$. *Right*: FID of trained SNGAN tickets found by IMP, Random Pruning, OMP, and Network Slimming [97].

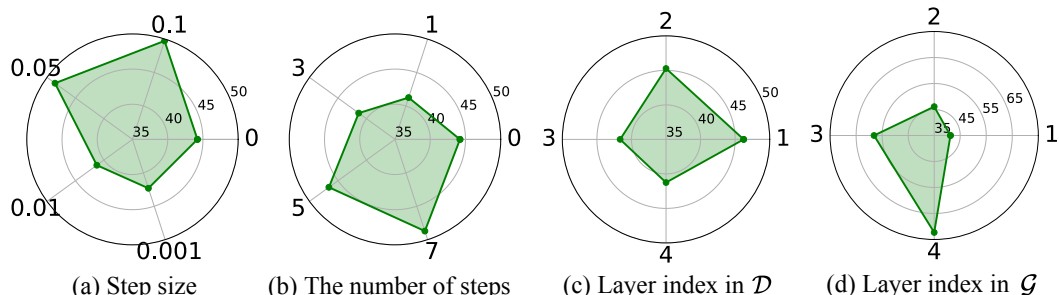

|(a) Step size | (b) The number of steps | (c) Layer index in $\mathcal{D}$ | (d) Layer index in $\mathcal{G}$ |

Figure 8: Ablation study on the location and strength of introducing AdvAug to data-efficient GAN training. The step size and the number of steps roughly indicate the strength of generated adversarial perturbations, e.g., a smaller step size or fewer steps for PGD means less aggressive perturbations [69]. FID ($\downarrow$) is reported.

## 4.4 Ablation and Analysis

**Pruning Ratio $\rho$ in the Ticket Finding.** To understand the effect of the pruning ratio in IMP to the quality of data-efficient GAN tickets, we experiment on SNGAN with 10% data of CIFAR-10 and $\rho = 10\%, 20\%, 40\%$ as the pruning ratio. As shown in Figure 7 (*Left*), all three IMP settings find data-efficient winning tickets in GAN; IMP with a lower pruning ratio tends to identify higher-quality GAN tickets in terms of FID, while it usually costs much more to reach the same level of sparsity as higher pruning ratios do.

**Augment $\mathcal{G}$ or $\mathcal{D}$: Either or Both.** We apply AdvAug on $\mathcal{G}$ or $\mathcal{D}$ only, and AdvAug on both $\mathcal{G}$ and $\mathcal{D}$. Only 10% data of CIFAR-10 are available for finding GAN tickets and training with AdvAug. Results are summarized in Figure 7 (*Middle*). Either employing AdvAug on $\mathcal{D}$ or $\mathcal{G}$ consistently obtains significant performance improvements (i.e., largely reducing the FID) over all sparsity levels, and augmenting both $\mathcal{D}$ and $\mathcal{G}$ further enhance the found data-efficient GAN tickets. Our results also show that sparser GAN tickets can benefit more from AdvAug, such as subnetworks with $20\%, 36\%, 83.22\%$ sparsity.

**Strength and Locations of Injecting AdvAug.** To better interpret the influence of the strength and layer locations of injected adversarial feature perturbations, we comprehensively examine SNGAN on 10% training data of CIFAR-10 across different step sizes, the number of PGD steps, and locations (i.e., where to apply AdvAug). When studying one of the factors, we fix the other factors with the best setup. From Figure 8, several observations can be drawn:

- Figure 8 (a) and (b) show that adopting AdvAug with step size 0.01/0.001 and PGD-1/3 assists the data-efficient GAN training, while AdvAug with step size 0.05/0.1 and PGD-5/7 perform worse than the baseline without AdvAug (i.e., the setting with zero step size or zero step PGD in Figure 8). It reveals that overly strong AdvAug can hurt performance.
- As shown in Figure 8 (c) and (d), augmenting the last layer of the discriminator $\mathcal{D}$ and the first layer (i.e., the closest layer to the latent input vector) of the generator $\mathcal{G}$ appears to be the best configuration for utilizing AdvAug. It seems that injecting adversarial perturbations into the "high-level" feature embeddings in general benefits more to mitigate the overfitting issue in data-limited regimes.

In summary, we observe that applying AdvAug to the last layer of $\mathcal{D}$ and the first layer of $\mathcal{G}$, with PGD-1 and step size 0.01, seems to be a sweet-point configuration for data-efficient GAN training, which is hence adopted as our default setting.

**Comparison with Baselines.** Naive baselines, i.e., random pruning, one-shot magnitude pruning (OMP) [87, 16], network slimming (NS [97], and random noise augmentation, are evaluated in Figure 7 (*Right*) and Table 6. Compared to random pruning, OMP and NS, IMP produces much better GAN tickets, especially at high sparsity levels (e.g., $\geq 48.8\%$). Compared to augmenting features with random noise sampled from $\mathcal{N}(0, 0.01^2)$, AdvAug also achieves larger performance gains on both $100\%$ and $10\%$ training data regimes.

Table 6: Performance of SNGAN models augmented by Gaussian Noise or AdvAug on $100\%$ and $10\%$ training data.

| Methods | 100% training data | | 10% training data | |
|---|---|---|---|---|
| | IS | FID | IS | FID |
| Baseline | 8.29 | 15.69 | 5.24 | 44.22 |
| + Gaussian Noise | 8.30 | 14.52 | 5.53 | 44.86 |
| + AdvAug | 8.42 | 13.99 | 6.10 | 41.25 |

## 5   Conclusion and Discussion of Broader Impact

We introduce a novel perspective for data-efficient GAN training by leveraging lottery tickets, which augmentations can further enhance, including our newly introduced feature-level augmentation. Comprehensive experiments consistently demonstrate the effectiveness of our proposal, on diverse GAN architectures, objectives, and datasets. Note that although finding lottery tickets requires a costly train-prune-retrain process, only *data efficiency* is of interest in this work. An intriguing future work would be to pursue data and resource efficiency (training and inference) together.

This research aims to enhance GAN training in the limited data regimes. However, it might amplify the existing societal risk of applying GANs. For example, the issue of image generation bias may be impacted or even amplified by the sparse structures, which we will verify in future work. The data-efficient generation ability might also be leveraged by undesired applications such as DeepFake.

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
