## A1 More Methodology Details

### A1.1 More about Feature-level Augmentation in GANs via Adversarial Training

**Adversarial feature-level augmentation on generator $\mathcal{G}$.** Denote the generator as $\mathcal{G} = \mathcal{G}_2 \circ \mathcal{G}_1$. Adversarial perturbations $\hat{\delta}$ generated by PGD, are applied to the intermediate feature space $\mathcal{G}_1(z)$, which can be depicted as follows:

$$\mathcal{L}_{\mathcal{G}}^{\mathrm{adv}} := \max_{\|\hat{\delta}\|_\infty \leq \epsilon} \mathbb{E}_{z \sim p(z)}[f_{\mathcal{G}}(-\mathcal{D}(\mathcal{G}_2(\mathcal{G}_1(z) + \hat{\delta})))], \tag{3}$$

$$\min_{\theta} \mathcal{L}_{\mathcal{G}} + \lambda_1 \cdot \mathcal{L}_{\mathcal{G}}^{\mathrm{adv}}, \tag{4}$$

where $\lambda_1$ controls the influence of adversarial information. We choose $\lambda_1 = 1$ tuned by a grid search.

**Adversarial feature-level augmentation on discriminator $\mathcal{D}$.** Denote the discriminator as $\mathcal{D} = \mathcal{D}_2 \circ \mathcal{D}_1$. We augment features of both real and generated samples. Specifically,

$$\mathcal{L}_{\mathcal{D}}^{\mathrm{adv}} := \min_{\|\delta\|_\infty \leq \epsilon} \mathbb{E}_{x \sim p_{\mathrm{data}}(x)}[f_{\mathcal{D}}(-\mathcal{D}_2(\mathcal{D}_1(x) + \delta))] +$$

$$\min_{\|\hat{\delta}\|_\infty \leq \epsilon} \mathbb{E}_{z \sim p(z)}[f_{\mathcal{D}}(\mathcal{D}_2(\mathcal{D}_1(\mathcal{G}(z)) + \hat{\delta}))], \tag{5}$$

$$\max_{\phi} \mathcal{L}_{\mathcal{D}} + \lambda_2 \cdot \mathcal{L}_{\mathcal{D}}^{\mathrm{adv}}, \tag{6}$$

where adversarial perturbations $\delta$ and $\hat{\delta}$ are applied to intermediate features $\mathcal{D}_1(x)$ and $\mathcal{D}_1(\mathcal{G}(z))$, respectively. $\lambda_2$ balances the effects of clean features and adversarial augmented features. In our case, $\lambda_2 = 1$ tuned by a grid search.

**The overall pipeline of AdvAug.** As presented in Figure 3, we augment the intermediate features of both the discriminator and generator. First, for augmenting $\mathcal{D}$, it minimizes $\mathcal{L}_{\mathcal{D}}^{\mathrm{adv}}$ to craft the adversarial perturbations for features from both real data and generated samples, and then maximizes $\mathcal{L}_{\mathcal{D}}$ together with $\mathcal{L}_{\mathcal{D}}^{\mathrm{adv}}$ to update the discriminator according to Eqn. 6. Augmenting $\mathcal{G}$ works similarly, but only on generated samples' features $\mathcal{G}_1(z)$. The full algorithm of training GAN with both **data- and feature-level augmentations** is summarized in Algorithm 2.

## A2 More Implementation Details

### A2.1 More Details of Adopted Datasets

**Complete descriptions.** The CIFAR-10 and CIFAR-100 datasets each consist of $60,000$ $32 \times 32$ color images in $10/100$ classes, with $6,000/600$ images per class, respectively. The ratio between the number of training and testing images is $5:1$. Tiny-ImageNet contains $200$ image classes, a training/validation/test dataset of $100,000/10,000/10,000$ $64 \times 64$ images. ImageNet has $1,000$ image classes, $1,281,167$ training samples, and $50,000$ validation samples. In all experiments, we use $128 \times 128$ resolution for ImageNet samples.

**Download links.** We list the download links for adopted datasets as follows:

($i$) CIFAR-10/100: https://www.cs.toronto.edu/~kriz/cifar.html

($ii$) Tiny-ImageNet: https://www.kaggle.com/c/tiny-imagenet

($iii$) ImageNet: http://www.image-net.org

($iv$) Few-shot datasets [1]: https://hanlab.mit.edu/projects/data-efficient-gans/datasets/

**Train-val-test splitting and subset constructions.** We follow the official splitting in the datasets. To construct subsets for the limited-data GAN training, we randomly sample a certain portion (e.g., $10\%$) from full training sets.

Table A7: FreezeD [94] results with/without our proposed training framework.

| Methods | 100-shot by [1] | | | AnimalFace | |
| --- | --- | --- | --- | --- | --- |
| | Obama | Grumpy Cat | Panda | Cat | Dog |
| FreezeD (0.00%) | 41.87 | 31.22 | 17.95 | 47.70 | 70.46 |
| FreezeD (0.00%) + DiffAug + AdvAug | 36.52 | 30.04 | 16.23 | 46.39 | 64.21 |
| FreezeD (48.80%) | 40.10 | 30.16 | 16.52 | 46.58 | 66.74 |
| FreezeD (48.80%) + DiffAug + AdvAug | 35.25 | 29.62 | 15.19 | 45.94 | 61.30 |

Table A8: Transfer performance of winning tickets found with FreezeD [94] and StyleGAN-V2 on FFHQ.

| Methods | 100-shot by [1] | | | AnimalFace | |
| --- | --- | --- | --- | --- | --- |
| | Obama | Grumpy Cat | Panda | Cat | Dog |
| StyleGAN-V2 finetune (0.00%) | 41.87 | 31.22 | 17.95 | 47.70 | 70.46 |
| StyleGAN-V2 finetune (0.00%) + DiffAug + AdvAug | 36.52 | 30.04 | 16.23 | 46.39 | 64.21 |
| StyleGAN-V2 finetune (48.80%) | 41.33 | 30.68 | 16.47 | 46.75 | 68.50 |
| StyleGAN-V2 finetune (48.80%) + DiffAug + AdvAug | 35.90 | 29.73 | 14.86 | 46.01 | 63.15 |

## A2.2 More Details of Reported Sparsity

**How is the sparsity level selected?** We perform iterative magnitude pruning which each time removes a fixed portion (e.g., $20\%$) of the remaining weights with the smallest magnitudes, leading to the series of sparsity levels like $\{20\% \ (1 - 1 \times 0.8), 36\% \ (1 - 1 \times 0.8^2), 49\% \ (1 - 1 \times 0.8^3), 59\% \ (1 - 1 \times 0.8^4), 67\% \ (1 - 1 \times 0.8^5)\}$. It is a widely adopted fashion in the literature [18, 16] of the lottery tickets hypothesis, and we strictly follow the standard convention.

## A3 More Experimental Results

**Comparisons with a smaller network baseline.** To show our achieved improvements not only come from the reduced network capacity but also from the sparse topology, we implement the "small-dense" baseline by shrinking the number of channels and constraining its number of parameters to be equivalent to that of the sparse subnetwork. We take $67.24\%$ sparse BigGAN on $10\%$ training data of CIFAR-100 as the experimental setup. Then we train them together with DiffAug and AdvAug, and report the (FID↓). LTH : Random Pruning : small-dense : Dense = $22.37 : 25.73 : 23.58 : 23.94$. The results indicate the small-dense baseline with reduced sample complexity is helpful ($23.58$ v.s. $23.94$), while most of the benefits come from the identified sparse structure of winning tickets ($22.37$ v.s. $23.94$). The sparse structure of subnetworks matters. Meanwhile, we notice that recent literature also share consistent findings: ($i$) big models are better few-shot learners [98]; ($ii$) big models produce better winning lottery tickets [59].

**Generalization study of our proposal.** Our framework is generalizable across diverse GAN architectures, which is also carefully evidenced in our main text (i.e., SNGAN, BigGAN, StyleGAN-v2). To further demonstrate it, we conduct extra experiments to combine our training framework with the proposed GAN architecture (i.e., + skip + decode) from [99]. We observe that sparse GAN tickets at $36\%$ sparsity with augmentations further obtain ($2.03, 0.75, 0.26$) FID reductions on (Obama, Grumpy cat, Panda), which again validates the effectiveness of our proposal.

**Pruning and augmentations on baseline pre-trained methods.** We apply our proposed training framework (LTH pruning + augmentations) to the baseline pre-trained method in Table 5. The performance of FreezeD with a pre-trained StypleGAN-V2 is collected in Table A7. We find consistent observations that our training framework (LTH pruning + augmentations) benefits FreezeD on few-shot generation tasks.

**Mask transferring.** As demonstrated in [62, 59], the winning tickets found on the pre-training task, show impressive transferability to diverse downstream tasks. We conduct similar pre-training and transfer studies in our context. Precisely, we first identify a "pre-training" GAN winning ticket with the FreezeD method [1] and the StyleGAN-V2 backbone on the FFHQ dataset. Then, we fine-tune it on diverse few-shot domains and report their performance in Table A8. We find that

Table A9: FID (↓) and IS (↑) results of SNGAN with 10% training data of CIFAR-10 at diverse sparsity levels. The setting "Sparse Tickets + Aug" is reported here.

| Sparsity | Dense (0%) | 5% | 10% | 15% | 20% | 25% | 30% | 35% | 40% | 45% | 50% |
|---|---|---|---|---|---|---|---|---|---|---|---|
| FID (↓) | 41.25 | 39.31 (↓1.94) | 36.85 (↓4.40) | 34.09 (↓7.16) | 32.47 (↓8.78) | 33.62 (↓7.63) | 36.38 (↓4.87) | 35.16 (↓6.09) | 35.94 (↓5.31) | 36.40 (↓4.85) | 37.22 (↓4.03) |
| IS (↑) | 5.64 | 5.72 (↑0.08) | 5.87 (↑0.23) | 6.03 (↑0.39) | 6.20 (↑0.56) | 6.14 (↑0.50) | 5.97 (↑0.33) | 5.93 (↑0.29) | 5.96 (↑0.32) | 5.91 (↑0.27) | 6.01 (↑0.37) |

in this practical and meaningful pre-training + fine-tuning scheme, our proposed LTH pruning + augmentations method is still effective.

**Fine-grained sparsity levels.** To demonstrate our proposal's effectiveness across diverse sparsity levels, we adjust the pruning ratios so that each time we remove 5% of the total weights with the smallest magnitudes, and conduct extra experiments on these sparsity levels {5%, 10%, 15%, 20%, 25%, 30%, 35%, 40%, 45%, 50%}. Results in Table A9, evidence the consistent benefits from our proposed training pipeline. All experimental configurations are the same as the ones in Figure 4.