# OpenReview forum: "Data-Efficient GAN Training Beyond (Just) Augmentations: A Lottery Ticket Perspective"
_NeurIPS.cc/2021/Conference — NeurIPS 2021 Poster_

### Official Review · Reviewer_Eqct · 2021-07-09

**Rating:** 7
**Confidence:** 4

**Summary:**

The paper addresses data-efficient GAN training.  It decomposes the task of the GAN training in low data regimes into two sub-problems: 1) finding the sparse subnetwork, so called lottery ticket, in the original GAN model, based on observations in [1,2], on a small set of training images; 2) then training the found sparse subnetwork, which is a more data-efficient solution in comparison to the dense one, on the same small training set.  In addition, the paper proposes a feature augmentation technique, which injects adversarial perturbations into intermediate features of both discriminator and generator. The paper shows that the sparse subnetwork can benefit from extra data- and feature-level augmentations.

**Ethical Concerns:**

No ethical concerns.

**Limitations And Societal Impact:**

The paper briefly discusses limitations and potential negative societal impact of the proposed approach.

**Main Review:**

Strengths:

-	The paper addresses a challenging and important problem of training GANs in low-data regimes. The extension of the lottery ticket hypothesis to training GANs in low data regimes is a novel and interesting idea.
-	The proposed feature augmentation helps to improve synthesis performance in both high and low data regimes for dense and sparse GAN networks. It’s also complementary to data augmentation techniques, such as ADA [3] or DiffAug[4].
-	The paper conducts experimental evaluation using different GAN architectures (BigGAN, SNGAN, StyleGAN-v2) across different datasets, showing the superior performance of sparse networks in comparison to their dense baselines in low data regimes.

Weaknesses:

-	It would be good to see the efficiency comparison between different sparse and dense networks during training and inference, e.g. what is the throughput of the networks? Number of parameters? Who long does it take to train the sparse subnetwork in comparison to its dense baseline?
-	In low-data regimes the difference between using the dense network and its sparse subnetwork in terms of FID is quite minor, ~1 FID point, e.g 13.59 vs. 13.05 FID on CIFAR100 in Table 4. It’s not obvious what is the benefit of the proposed approach as the performance improvement is marginal and finding lottery tickets requires costly train-prune-retrain process. Thus the paper would benefit from the efficiency analysis (see the point above).
-	It’s not clear form the paper if in low data regime a smaller network would perform better in comparison to the proposed method. A simple comparison of the found sparse subnetwork to the dense baseline with the reduced number of parameter (e.g. number of channels), equivalent to the number of parameters of the sparse subnetwork, would be interesting to see. It might happen that with little amount of training data a smaller network (in terms of parameters) will help to overcome overfitting and might perform even better.
-	The paper misses comparison with the related work on few-shot generation task: Liu et al., “Towards faster and stabilized gan training for high- fidelity few-shot image synthesis”, ICLR, 2021. This work reaches superior performance in comparison to the proposed approach in terms of FID, e.g. on Obama dataset 52.86 FID in Table 5 vs. 41.05 FID in Liu et al., on Panda 14.75 vs. 10.03 FID, on Animal Face – Cat 47.40 vs. 35.11 FID. Note that both approaches do not use any pre-training, thus the comparison is fair.

Typos: Line 164: Figure 2 -> Figure 1, Table 3 “10.14” should not be in bold.


**Time Spent Reviewing:**

3h

---

> ### Author Response · Authors · 2021-08-10
> **Response to Reviewer Eqct [Cons1-5]**
>
> Thanks for rating our paper as interesting and novel. We provide point-wise responses to your constructive comments as below.
>
> **[Cons1. Efficiency comparison.]** As mentioned in the abstract and line 319, this work is focused on data efficiency. Our current implementation took roughly the same overall (re-)training time as unpruned baselines. It is possible to boost data and resource efficiency (e.g., time) together – we take it as future work. In terms of the inference efficiency, for a GAN winning ticket at 67.24% sparsity, it can save ~60-63% parameters and ~55%-60% inference FLOPs.
>
> **[Cons2. Marginal improvement and costful finding.]** We respectfully disagree. First, we think the costful finding of LTH is not a concern for this paper, since i) **we focus on improving data efficiency**; and ii) it is not costful in the inference stage, trained GANs still enjoy the extra bonus of reduced parameters and inference FLOPs.
> Second, as nicely summarized by Reviewer npsk, “The results are very strong across various datasets, some latest GAN models”. We argue as follows:
> - We think the not-large performance gap in Table 4 is mainly because the performance (FID:12.11) of StyleGAN-v2 is almost saturated with 10% training data on the simple CIFAR-100 data. Note that on more complicated image datasets, our methods obtain **11.3 FID reduction** in 10% training data of Tiny-ImageNet (Table 1), and up to **6.62 FID reduction** on few-shot generation in Table 5. Our performance gains are substantial, agreed by Reviewer npsk as “the results are very strong”.
> - Our obtained improvements are consistent across different GAN architectures, datasets, and augmentations.
> - Although there is ~1 FID point gain in CIFAR datasets, it is statistically significant due to the non-overlapped error bars.
>
> **[Cons3. The smaller network baseline.]** Thanks for the great suggestions. To address your concerns, we implement the “small-dense” baseline by shrinking the number of channels and constraining its number of parameters to be equivalent to that of the sparse subnetwork. We take 67.24% sparse BigGAN on 10% training data of CIFAR-100 as the experimental setup. Then we train them together with DiffAug and AdvAug, and report the (FID↓). LTH : Random Pruning : **small-dense** : Dense = 22.37 : 25.73 : 23.58 : 23.94. The results indicate the small-dense baseline with reduced sample complexity is helpful (23.58 v.s. 23.94), while most of the benefits come from the identified sparse structure of winning tickets (22.37 v.s. 23.94). The sparse structure of subnetworks matters. Please also kindly check the answer to Reviewer npsk’s Cons3&4.
>
> Meanwhile, we notice that recent literature also share consistent findings:
> - Big models are better few-shot learners (NeurIPS’20 - Big Self-Supervised Models are Strong Semi-Supervised Learners);
> - Big models produce better winning lottery tickets (CVPR’21 - The Lottery Tickets Hypothesis for Supervised and Self-supervised Pre-training in Computer Vision Models).
>
> **[Cons4. Missed related work.]** Thanks for pointing out the related work. We think our proposed training framework works orthogonally with Liu’s ICLR paper, and can be combined to deliver better few-shot results. Our framework is generalizable across diverse GAN architectures, which is also carefully evidenced in our paper (SNGAN, BigGAN, StyleGAN-v2), as also recognized by Reviewer npsk.
>
> To further address your concerns, we conduct extra experiments to combine our training framework with the proposed GAN architecture (i.e., +skip+decode) from Liu’s ICLR21 paper. We observe that sparse GAN tickets at 36% sparsity with augmentations further obtain (2.03,0.75,0.26) FID reductions on (Obama, Grumpy cat, Panda), which again validate the effectiveness of our proposal. More results, citations, and related analyses will be included in our final version.
>
> **[Cons5. Typo]** Thanks for such detailed comments. We will perform more proofreading and address the typos and wrong bold fonts in the revision.

---

> > ### Comment · Reviewer_Eqct · 2021-08-18
> > **Response to rebuttal**
> >
> > Thanks a lot for your detailed feedback and all clarifications! The provided extra experiments look quit convincing, showing consistent improvement across different settings. I think adding the efficiency analysis as well as comparison with Liu et al. ICLR work to the paper would be beneficial to the readers and would make the paper stronger.
> >
> > Assuming that all the stated changes are made to the final version, I would be happy to accept this work.

---

> > > ### Author Response · Authors · 2021-08-18
> > > **Response to Reviewer Eqct**
> > >
> > > Dear Reviewer Eqct,
> > >
> > > Many thanks for reviewer Eqct's constructive review, and the acknowledgment of our efforts in the rebuttal. We are glad that our response has solved the concern. We promise that all stated changes in our rebuttal will be delivered in our final version, especially for the efficiency analysis and the comparison with Liu et al. ICLR.
> > >
> > > Given the current opinions, we humbly ask if reviewer Eqct could take the time to increase the score, which might more accurately reflect your positive assessment of our work. We are again very thankful for your time and support!
> > >
> > > Best wishes,
> > >
> > > Authors

---

### Official Review · Reviewer_npsk · 2021-07-16

**Rating:** 8
**Confidence:** 5

**Summary:**

Inspired by the lottery ticket observation, the authors decomposed GAN training into two stages: (i) finding sparse winning tickets in GANs by iterative pruning; (ii) training the found sparse tickets with various augmentations. Either sub-problem becomes more data-efficient to train, and they can re-use the same small set of real images.

**Limitations And Societal Impact:**

It’ll be great if the authors can address the following concerns:

1. Table 1, Why 100% data was hurt by sparsity 67.24%, with or without AdvAug? Similarly in Table 3 CIFAR-10, Dense BigGAN outperformed sparse in all three conditions. It’s a bit confusing as the benefit of sparsity does not show to be fully consistent.

2. Table 4 StyleGAN-v2: neither AdvAug, nor DiffAug + AdvAug was reported, This looks weird, since previous BigGAN and SNGAN experiments mainly reported on the two.

3. Sparsity during training seems to benefit data-efficient learning, and LTH is one of the possible sparse training ways. Have you tried other ideas, such as SNIP or dynamic sparse training?

4. I can understand sparsity might constrain the hypothesis space and reduce the sample complexity. But why the sparsity structure pattern also matters in data efficiency (e.g., LTH largely outperforms random pruning)?

5. I don’t quite understand why network slimming performs so badly in Figure 7.

6. How about this algorithm applied to image-to-image translation GANs? I think this would be as interesting, such as developing some few-shot super-resolution models in some data-scarce scientific imaging domains.

7. Typos are non-sparse in the paper: “mini-max”, “and we then training them…”, and more. Proofreading is needed.


**Main Review:**

This paper is a very strong empirical work on data-efficient GAN training. To my best knowledge, the authors are the first to show these sparse networks are much more efficient to train with smaller sets of data than their dense counterparts, and hence provide a blessing for few-shot regimes.

From a fundamental perspective, it goes in parallel with the existing LTH literature along with the inference efficiency of networks. Although the lottery ticket phenomenon was discovered by other prior works, no existing work ever considered the influence of training data size on locating and training those tickets in general. It will inspire more re-thinking on the value of network sparsity, for not only GANs but also general deep NNs.

The results are very strong across various datasets, some latest GAN models (including BigGAN and StyleGAN-V2), and the metrics reported.  Ablation studies are thorough and clear to follow.  In particular, the authors compared the two state-of-the-art methods DiffAug and ADA, and they were even able to include the very new RLC method (CVPR’21) into comparison, as well as their various combinations. I salute to the authors’ extensive efforts.

The authors’ technical approach is quite straightforward. But I feel to some extent, it is a pro rather than a con. Given the new angle (LTH for data-efficient training) as the main delivery of this paper, I like the authors could exploit this insight with very easy-to-follow approaches, and I value a simple method that works while bringing in new insights.

The writing clarity is good, but the AdvAug part seems a bit disconnected from the main theme of LTH.


**Time Spent Reviewing:**

4

---

> ### Author Response · Authors · 2021-08-10
> **Response to Reviewer npsk [Cons1-7]**
>
> Many thanks for the detailed summary and positive comments. We’re very glad you rate our work as novel and strong, and likewise, we found the set of perceptive questions you raised in your feedback very insightful, pushing us to further improve our paper.
>
> **[Cons1. 100% data was hurt by sparsity 67.24%?]** Fitting 100% data well potentially requires GAN with larger remaining capacity (e.g., more parameters or smaller sparsity). This is why the aggressive 67.24% sparsity will hurt the performance on 100% data GAN training.  Note that we intentionally take 100% training data as our data-rich baseline (not our target data-limited setup) to positively support that data-limited GAN training (e.g., 10%, 20%) obtains substantial gains from sparsities. Specifically, in Table 1, for data-limited setups (our paper’s main focus), both sparse structural priors from the lottery ticket pruning and augmentations boost GAN’s performance. In summary, 100% data training tends to benefit from smaller sparsity (e.g., 36% sparsity in Table 1) and limited data (20%,10%) training tends to benefit from larger sparsity (67.24% in Table 1&3). More analyses are in line 220.
>
> **[Cons2. Table 4 needs AdvAug or DiffAug + AdvAug]** Thanks for pointing it out. As you suggested, we conduct extra experiments on sparse StyleGAN-V2 tickets at 48.80% sparsity and 10% training data of CIFAR-100. Results (FID↓) are collected as below: (a) + AdvAug, 12.61 +- 0.04; (b) + DiffAug + AdvAug, 12.27 +- 0.03. The improvements are consistent with our other experiments.
>
> **[Cons3. Other pruning methods]** We agree that there are other possible ways to explore sparsities that may also benefit data-efficient GAN training. As you suggested, we implement SNIP and random pruning to locate 67.24% sparse BigGAN on 10% training data of CIFAR-100. Then we train them together with DiffAug and AdvAug, and report the (FID↓). LTH : SNIP : Random Pruning : Dense = 22.37 : 24.01 : 25.73 : 23.94. We observe that pruning methods (or sparse masks’ quality) matter, and randomly removing weights hurts the GAN performance, which aligns with the findings in [2]. We will explore more pruning methods including dynamic sparse training in the future.
>
> **[Cons4. Why is LTH better than RP?]** The LTH literature [2,18] shows that LTH pruning can locate critical sparse connectivities which have unimpaired or even surpassed trainability and generalization compared to dense models, where random pruning cannot make it. And these general benefits have been broadly demonstrated in diverse applications [18, 45, 61, 57]. Meanwhile, LTH explores and enforces good inductive bias [ref. a&b] (e.g., enlarged convex region near the optimal [ref.a]). It can be utilized as a sparse structural prior to alleviate the data-hungry issue in our data-limited GAN training, which also explains the superiority of LTH compared to RP.
>
> [ref.a] Why Lottery Tickets Wins? A Theoretical Perspective of Sample Complexity on Sparse Neural Networks?
>
> [ref.b] Sifting out the features by pruning: Are convolutional networks the winning lottery ticket of fully connected ones?
>
> **[Cons5. Why is network slimming bad?]** Network slimming here is a structural pruning method that removes certain whole channels with respect to the importance scores. Such aggressive structural pruning usually performs worse than unstructured pruning (LTH), as also evidenced in [2].
>
> **[Cons6. Image-to-image translation?]** Thanks for the great and interesting suggestion. It is worthy to mention that our proposed framework is independent of GAN’s architecture and task. Also, based on GAN tickets’ results of image-to-image translation in [2], we believe our methods can also generalize well and be effective in image-to-image translation tasks. Due to the limited time in the rebuttal period, we promise these results in our final version.
>
> **[Cons7. Typo]** Thanks so much for such detailed comments. We will perform more proofreading and address all the typos in the revision.

---

### Official Review · Reviewer_6Tg4 · 2021-07-25

**Rating:** 7
**Confidence:** 4

**Summary:**

The authors propose using iterative magnitude pruning, along with two novel augmentation strategies as a means to achieve few-shot GAN training.


**Limitations And Societal Impact:**

Yes, they have, albeit in a clumsy manner, as I explain in the Details/Conclusion section above.


**Main Review:**

Novelty and research direction:

Finding ways to train GANs data-efficiently is a very intriguing premise. Attempting to use the lottery ticket hypotheses insights into achieving this is a very interesting research direction. The work uses previous work relating to ITP on GANs, on the few-shot domain, which is a fairly novel direction.

--------------------------------------------------------------------------------------------------------------------------------------------------------------------------------------
Writing quality:

Overall the paper’s quality is decent but suffers from clumsy writing in various places, as well as some mild lack of conciseness and precision.

--------------------------------------------------------------------------------------------------------------------------------------------------------------------------------------
Summary of review:

The idea proposed is novel and very useful, however, the experiments undertaken do not sufficiently disentangle key variables that are necessary to draw a useful conclusion. Find the full details in the details section below (the experiments section is the one with the most value).

Furthermore, the writing quality of the paper is at times clumsy, as well as mildly imprecise and inconcise.

I cannot recommend this paper, in its current state for acceptance.

--------------------------------------------------------------------------------------------------------------------------------------------------------------------------------------

Details:

1. Abstract:
The abstract is too large, and suffers from precision and clarity issues. Furthermore, the usage of citations in an abstractly is generally not recommended, at least in ML. Ideally an abstract should be 5-8 sentences long, unless there are special reasons for not doing so.

2. Introduction:

The very first sentence seems to be lacking precision since you imply that all modern advancements in GANs can be attributed to better data and nothing else. Also the usage of the verb ‘blessed’ here feels a bit out of place, but that’s just my taste. The same term is reused in line 55, which again feels out of place.

The rest of the introduction seems to suffer from clumsy writing in a few places, but overall flows well and distills down the main idea well, but fails to clearly nail down the exact contributions the authors are claiming.

3. Related Work:

Very good related work section, that seems to have a few places of clumsy writing, but overall seems to be on point.

Also, some notable papers relating to few-shot learning + GANs seem to be missing, i.e.

 i. https://arxiv.org/abs/1711.04340

 ii. https://arxiv.org/abs/1711.04043

4. Methodology:

AdvAug description is inadequate, how is PGD applied? Optimized against what? The figure provides no information, neither does Equation 1. This is a majour point, and should not be in the appendix. I was able to figure out what was going on, but had majour trouble piecing it together without the appendix.
The rest of the section seems to be OK. However, the last sentence in 3.2 does not make much sense to me, since it’s overly convoluted.

5. Experiments:

Overall, the experiments conducted to showcase the usefulness of advAug were very interesting.

That being said, I have some reservations regarding the experiments in Table 1 and 2. Is your pruning method causing the improved performance there? Or is it the mere fact that you are using fewer weights, which happen to create a regularization effect that produces better generalizing generators? For Table 1 to be better able to showcase your point, you need to run experiments where you randomly remove 36% and 67.24% of all weights and train on all the various data settings. This way you can see if your proposed methods offer any benefit, or whether it’s all due to a smaller model with what is probably a pretty tree-esque structure, forcing bottlenecks left and right, and inadvertently causing regularization.

Furthermore, the experiments conducted in Table 5 seem a bit unfair. It seems that most benefit comes from the augmentations you used. Have you tried applying those to the baseline pretrained methods? That would tell you if your pruning method is useful or not. Furthermore, again, just like in my previous point, you need to ensure that any added performance comes from your pruning method and not from the reduced model size. You can verify that by randomly dropping say 48.80% of your weights, by masking them away, and training the rest of the model to compare a random pruning vs the guided pruning you are already showcasing in the paper.

Furthermore, it would have been cool to see how finding a lottery ticket GAN on the pretraining dataset, would transfer to the few-shot tasks. That could also push your model performance further.

Overall, the experiments don’t do a good enough job at disentangling some important factors required to draw useful conclusions about the claims made by the authors.

6. Conclusion:

Line 322-324 states that the work is scientific in nature and that this somehow means that the work will not pose substantial risk of societal harm. I find this statement a bit odd. Does work that is scientific get a risk-free card? Isn’t science a very powerful tool that can both protect and destroy? I am a bit confused here. Did the authors mean that the work is far removed from a practical application that could be malicious because it is highly experimental? Please clarify.

--------------------------------------------------------------------------------------------------------------------------------------------------------------------------------------

Improving the quality of this bar, to potentially meet my subjective NeurIPS bar

The work itself is interesting and novel. However, a few things need to be done to get this paper at a level where I’d be comfortable recommending acceptance.

1. Proof read the paper, and ensure the abstract is smaller (5-8 sentences), and make clarity and precision all around a priority. I find it useful to have colleagues from other institutions/countries review the work, as they are more objective and further removed from the work, that they can easily pick up any clarity/understanding issues.
2. For some very useful tips have a look at the below websites (which I use myself and my students all the time, and it’s really good stuff):

 a. https://cs.stanford.edu/people/widom/paper-writing.html

 b. https://www.easterbrook.ca/steve/2010/01/how-to-write-a-scientific-abstract-in-six-easy-steps/

3. Ensure your introduction has clearly outlined contributions, perhaps use a bullet list to make them stand out to the reader even more.
4. Run experiments with randomly removed weights, to confirm that any gains come from your lottery ticket hypothesis method and not just a reduced model size and weird bottlenecks that inadvertently pop up in pruned networks.
5. Run experiments with your proposed augmentations applied to your baseline methods, this way you can ensure that your proposed LTH method + augmentations produce the best results, or to at least learn whether your gains are all due to the augmentation strategies used.
6. Try to find a LTH GAN on the pretraining datasets of other transfer methods, then fine tune on those with additional LTH steps on the few-shot domains. This might also help your method by leveraging data that other methods already use.

--------------------------------------------------------------------------------------------------------------------------------------------------------------------------------------

**Time Spent Reviewing:**

2

---

> ### Author Response · Authors · 2021-08-10
> **Response to Reviewer 6Tg4 [Cons1-4]**
>
> Thanks for rating our work as **novel & useful**, and likewise, we found the perceptive writing issues you raised are very constructive and extremely helpful, pushing us to further improve our writing quality and readability. We point-wisely address all the mentioned writing issues in Cons1.
>
> **[Cons1. Paper Writing.]** Thanks again for such detailed writing suggestions.
> - *Abstract*: we will remove all citations, shorten the abstract to 5-8 sentences and improve its clarity. Here is our updated version with 5 sentences: “Training generative adversarial networks (GANs) with limited real image data generally results in deteriorated performance and collapsed models. To conquer this challenge, we are inspired by the latest observation, that one can discover independently trainable and highly sparse subnetworks (a.k.a., lottery tickets) from GANs. Treating this as an inductive prior, we suggest a brand-new angle towards data-efficient GAN training: by first identifying the lottery ticket from the original GAN using the small training set of real images; and then focusing on training that sparse subnetwork by re-using the same set. We find our coordinated framework to offer orthogonal gains to existing real image data augmentation methods, and we additionally present a new feature-level augmentation that can be applied together with them. Comprehensive experiments endorse the effectiveness of our proposed framework, across various GAN architectures (SNGAN, BigGAN, and StyleGAN-V2) and diverse datasets (CIFAR-10, CIFAR-100, Tiny-ImageNet, ImageNet, and multiple few-shot generation datasets).”
> - *Introduction*: we will remove the verb “blessed” and change the first sentence to “The quantity, diversity and high quality of natural images available in the general domain have played an important role in the breakthroughs achieved by Generative Adversarial Networks (GANs) over the past few years”. Meanwhile, we improve the clarity of the introduction and use a bullet list to present outlined contributions.
> Related works: thanks so much for pointing out the related reference of few-shot learning in GAN. We will add these citations in our revision.
> - *Methodology*: we will reorganize the descriptions of AdvAug by moving all necessary details (e.g., formulations, implementations, and algorithms) to the main text and making it self-contained. We will also rewrite the last sentence in Section 3.2 to “When the training data volume reduces from 100% to 10% of the full training set, the quality of sparse mask remains to be stable, since it can achieve matched performance compared to its dense counterpart in both full and limited data re-training regimes.”
> - *Experiment*: thanks for these constructive suggestions of the experiment design. All three experimental concerns of “randomly removing weights”, “augmentation on baseline methods”, and “pre-training and transferring” are addressed in the answers below (Cons 3, 4 & 5).
> - *Conclusion*: We apologize for our insufficient statements of negative social impact. We will restate it: “This research aims to enhance GAN training in the limited data regimes, and it is highly experimental. However, it might amplify the existing societal risk of applying GANs. For example, the issue of image generation bias may be impacted or even enlarged by the sparse structures. The data-efficient generation ability might also be leveraged by undesired applications such as DeepFake, which we need combat.”
>
> Thanks for all the writing suggestions. We will invite other researchers to have reviews about this work to further improve its writing clarity. Also, many thanks for sharing the websites about writing instructions, which is very useful for us.
>
> **[Cons2. Randomly removing weights.]** We in fact already reported the results of randomly pruned sparse networks in Figure 7 (right), as noticed by Reviewer npsk. It supports that the performance benefits from the sparse connectivities identified by LTH, not only from reduced model size. Please kindly check that.
>
> To further address your concerns, during rebuttal we (1) conduct more random pruning experiments; and (2) implement a “small-dense” baseline by shrinking the number of channels and constraining its number of parameters to be equivalent to that of the sparse subnetwork. We take 67.24% sparse BigGAN on 10% training data of CIFAR-100 as the experimental setup. Then we train them together with DiffAug and AdvAug, and report the (FID↓). LTH : Random Pruning : small-dense : Dense = 22.37 : 25.73 : 23.58 : 23.94. The results indicate the small-dense baseline with reduced sample complexity is helpful (23.58 v.s. 23.94), while most of the performance gains come from the identified sparse structure of winning tickets (22.37 v.s. 23.94). The sparse structure of subnetworks matters. Please also kindly check the answer to Reviewer npsk’s Cons3 for more comparison with other pruning methods.
>
> **[Cons3. Augmentations on baseline methods and a bit unfair in Table 5.]** Thanks for helping us to present a more rigorous experimental design. As you suggested, we apply our proposed framework (LTH pruning + augmentation) to the baseline pre-trained method in Table 5. The performance of FreezeD with a pre-trained StypleGAN-V2 is collected in the table below, and more results will be provided in our revision. We find consistent observations that our training framework (LTH+augmentation) benefits FreezeD on few-shot generation tasks.
>
> |Method|Obama|Grumpy Cat|Panda|AnimalFace-Cat|AnimalFace-Dog|
> |:-|:-:|:-:|:-:|:-:|:-:|
> |FreezeD (0.00%)|41.87|31.22|17.95|47.70|70.46|
> |FreezeD (0.00%)+DiffAug+AdvAug|36.52|30.04|16.23|46.39|64.21|
> |FreezeD (48.80%)|40.10|30.16|16.52|46.58|66.74|
> |FreezeD (48.80%)+DiffAug+AdvAug|35.25|29.62|15.19|45.94|61.30|
>
>
> Meanwhile, we see the comparison in Table 5 reasonable since we plan to show that our found data-efficient GAN tickets are capable of being trained from scratch (without any pre-training) and show competitive performance to other transfer learning algorithms on the few-shot generation task.
>
> **[Cons4. Pre-training and transfer.]** Thanks for the excellent suggestion of pre-training and transfer studies. As you suggested, we first identify a “pre-training” GAN winning ticket with the FreezeD method (i.e., one classical transfer method) and the StyleGAN2 backbone on the FFHQ dataset. Then, we fine-tune it on diverse few-shot domains and report their performance below. We find that in this practical and meaningful pre-training + fine-tuning scheme, our proposed LTH+augmentations method is still effective. Similar observations are also shared in recent LTH literature in both natural language processing and computer vision domains (e.g., a. The Lottery Ticket Hypothesis for Pre-trained BERT Networks; b. The Lottery Tickets Hypothesis for Supervised and Self-supervised Pre-training in Computer Vision Models).
>
> |Method|Obama|Grumpy Cat|Panda|AnimalFace-Cat|AnimalFace-Dog|
> |:-|:-:|:-:|:-:|:-:|:-:|
> |StyleGAN-V2 finetune (0.00%)|41.87|31.22|17.95|47.70|70.46|
> |StyleGAN-V2 finetune (0.00%)+DiffAug+AdvAug|36.52|30.04|16.23|46.39|64.21|
> |StyleGAN-V2 finetune (48.80%)|41.33|30.68|16.47|46.75|68.50|
> |StyleGAN-V2 finetune (48.80%)+DiffAug+AdvAug|35.90|29.73|14.86|46.01|63.15|

---

> > ### Comment · Reviewer_6Tg4 · 2021-08-18
> > **Response to rebuttal**
> >
> > Thanks for taking the time to read my review and make so many changes! I am glad you found my review useful. Sorry if it seemed a bit harsh, I was just trying to help make your work better, at least from my perspective. :)
> >
> > The new abstract is much better indeed.
> > Also, yes, considerations 2, 3 and 4 have now been met. Those results should help you draw more robust conclusions. Sorry for missing the table on randomly removing weights -- it might be worth making it clearer that those results are there in your paper, by better emphasising it.
> >
> > Assuming all the stated changes are made (writing-wise), I would be happy to accept this paper.

---

> > > ### Author Response · Authors · 2021-08-18
> > > **Response to Reviewer 6Tg4**
> > >
> > > Dear Reviewer 6Tg4,
> > >
> > > Many thanks for reviewer 6Tg4's constructive review, and the acknowledgment of our efforts in the rebuttal. We are glad that our response has solved the concern. We promise that all stated changes in our rebuttal will be delivered in our final version, including all new results, all modified sentences (writing-wise), and the highlight of the results of randomly removing weight.
> > >
> > > Given the current opinions, we humbly ask if reviewer 6Tg4 could take the time to increase the score, which might more accurately reflect your positive assessment of our work. We are again very thankful for your time and support!
> > >
> > > Best wishes,
> > >
> > > Authors

---

### Official Review · Reviewer_3bu4 · 2021-07-26

**Rating:** 6
**Confidence:** 4

**Summary:**

This paper proposed to extend Lottery Ticket Hypothesis (LTH) for data-efficient GAN training. Specifically, the authors have found that sparse subnetworks lead to better sample qualities of Image GANs. In addition, they have proposed AdvAug that combines the ideas of feature augmentation and adversarial training.

**Limitations And Societal Impact:**

Limitations or potential negative societal impact are not discussed in depth in this paper. Including limitations as well as the position of this paper against prior literature will greatly enhance the quality of this paper.

**Main Review:**

[Strengths]
- The paper is largely well-written and easy to understand.
- Using sparse subnetwork to enhance data efficiency seems novel (to the best of my knowledge).

[Weaknesses]
- Limited novelty in terms of method. This paper does not present any new techniques in the method section. Lottery ticket hypothesis was explored in prior literature. Also, using adversarial training to improve robustness of deep networks are pretty well-known in computer vision.
- Unconvincing experiment results. Since the novelty of method is limited, the value of the paper hinges on lessons we can learn from the experiments. From Table 1, on 100% and 20% data, sparse network doesn't seem to be helping at all. However, on 10% data, sparse subnetworks are helpful. The results are not convincing to me and seem to be coincidental. I believe there is a lot of randomness going on here. The results are averaged over 3 "evaluation" runs, but not on 3 "training" runs. I suggest running experiments on 3 different 10%/90% splits for example. Also, in Tables such as Table 4, gaps between different entries are not big enough. Although there is no overlap between error bars, I believe these error bars are related to different evaluation runs (not training runs). From my experience, this kind of numerical difference can be easily reversed if another trial of GAN training is performed.
- No insight regarding why sparse subnetwork helps data-efficiency of GAN training. Also, does this finding can be extended to image recognition benchmarks?

[Typo]
- In line 164: Fig 2 outer circle should be Fig 1 outer circle

**Time Spent Reviewing:**

1.5

---

> ### Author Response · Authors · 2021-08-10
> **Response to Reviewer 3bu4 [Cons1-5]**
>
> We politely yet firmly point out that your reviews seem to significantly misread our result and problem setting. We apologize if that was caused by any presentation or clarity issue. However, we think your questions have been all addressed below. We especially hope you pay more attention to all the latest results in Table 1.
>
> **[Cons1. Limited novelty?]** We respectfully disagree. What we really focus on advocating is the novel and practical angle of bridging LTH with data-efficient training for the first time. Uniquely and strongly motivated by this fresh angle, our solution integrates LTH and data augmentations – apparently, this was not naively “adopted from previous work”.
>
> Our novelty is consistently recognized by all three other reviewers. Specifically, please check **Reviewer npsk**’s nice summarization on our behalf: “Although the lottery ticket phenomenon was discovered by other prior works, **no existing work ever** considered the influence of training data size on locating and training those tickets in general. It will inspire more re-thinking on the value of network sparsity, for not only GANs but also general deep NNs.” Our novelty is also highly acknowledged by **Reviewer 6Tg4** as “intriguing premise”, “novel direction” and “novel and very useful”; by **Reviewer Eqct** as “a novel and interesting idea”. ***Also, using adversarial training in our paper is to improve generalization rather than adversarial robustness, which is less explored in the literature***.
>
> **[Cons2. Unconvincing experiment results?]** We respectfully argue that  the results of our experiments are strong, as also commented by Reviewer npsk as “the results are **very strong** across various datasets, some latest GAN models (including BigGAN and StyleGAN-V2)”. We provide pointwise responses to your concerns below.
> - **For results in Table 1, we’re really confused by your question which seems to refer to non-existing information (or perhaps due to significant mis-reading of our latest results)** In Table1, for 20% and 10% training data schemes, all reported numbers positively show clear performance improvements, which indicates sparse networks at both two sparsity levels benefit the data-limited GAN training. Also, for 100% training data, sparse GAN at 36% sparsity still obtains clear performance improvements. Furthermore, we intentionally show the impaired results of GAN at 67.24% sparsity with 100% training data as the data-rich baseline setting, not our focused data-limited setup. It actually positively supported that data-limited GAN training (e.g., 10%, 20%) obtains substantial gains from sparsities. More analyses are referred to line 220. **Therefore, we regret to say that your comments are factually flawed and confusing**.
> - The adopted setting of “average multiple evaluation runs” is strictly following the related seminal NeurIPS’20 paper - “Differentiable Augmentation for Data-Efficient GAN Training” (e.g., their Table 2).
> - We think the not-large performance gap in Table 4 is mainly because the performance (FID:12.11) of StyleGAN-v2 is almost saturated with 10% training data on the simple CIFAR-100 data. Note that on more complicated image datasets, our methods obtain **11.3 FID reduction** in 10% training data of Tiny-ImageNet (Table 1), and up to **6.62 FID reduction** on few-shot generation in Table 5. Our performance gains are substantial, agreed by Reviewer npsk as “the results are very strong”.
> - To further address your concerns, we repeat 3 training runs on different data splits as you suggested. We take 67.24% sparse BigGAN and dense BigGAN on 10% training data of CIFAR-100 as the experimental setup. Then we train them together with DiffAug and AdvAug, and report the FID scores: Dense BigGAN (24.01 +- 0.12) v.s. Sparse BigGAN (22.35 +- 0.16). Based on the new results, our improvements are statistically significant.
>
> **[Cons3. No insight of why sparsity helps?]** The insights of appropriate sparsity improving data efficiency lie in the following three aspects: (a) sparsity as a regularization could reduce overfitting, as is well known in classical ML [1-5]; (2) the lottery ticket, as a special sparse neural network, is recently found to reduce sample complexity in theory and practice. For example, [b] shows the number of samples required for achieving zero generalization error is proportional to the number of the non-pruned weights in the hidden layer. That was also echoed by Reviewer npsk commenting: “sparsity might constrain the hypothesis space and reduce the sample complexity”. (c) Sparsity is an useful inductive bias and prior knowledge for modeling images, which remains relevant in LTH/deep network domains, e.g. see [7]; (d) the unimpaired trainability [8] of lottery ticket further ensures the above benefits can be attained by standard SGD training from scratch.
>
> We hope the above explanations (a)-(d) can now convince you why sparsity/LTH is the rational way to go in our setting. We will be happy to discuss further during the rolling discussion. We plan to add the above explanations into the final paper. Besides, we also see other reviewers unanimously comment our angle as “strong” (Reviewer npsk), “important”, “novel and very useful” (Reviewer 6Tg4), “novel and interesting” (Reviewer Eqct).
>
> [1] Sparseout: Controlling Sparsity in Deep Networks
>
> [2] Understanding machine learning: From theory to algorithms
>
> [3] Occam’s razor. (NeurIPS 2001)
>
> [4] Non-vacuous Generalization Bounds at the ImageNet Scale: A PAC-Bayesian Compression Approach
>
> [5] Stronger Generalization Bounds for Deep Nets via a Compression Approach
>
> [6] Why Lottery Tickets Wins? A Theoretical Perspective of Sample Complexity on Sparse Neural Networks
>
> [7] Sifting out the features by pruning: Are convolutional networks the winning lottery ticket of fully connected ones?
>
> [8] The Lottery Ticket Hypothesis: Finding Sparse, Trainable Neural Networks
>
> **[Cons4. Generalizable to image classification?]** Although this paper focuses on tackling the data-efficient training of GANs, we believe that such a coordinated framework might potentially be generalized to training other deep models on other tasks (e.g., image recognition) with high data efficiency too. It is also highly acknowledged by Reviewer npsk as “it will inspire more re-thinking on the value of network sparsity, for not only GANs but also general deep NNs”.
>
> Furthermore, we have indeed conducted several experiments on image recognition in our ongoing project. We find the performance of a sparse ResNet-56 at 59% sparsity with 40%-50% training data of CIFAR-100 and proper augmentations, can almost match the performance of full data trained dense ResNet-56. Note that this is totally out of this paper’s scope, and we believe the generalization of our proposed framework beyond GAN training is an extra bonus, but not an essential component or any weakness for this current work.
>
> **[Cons5. Typo.]** Thanks for your detailed comments. We will perform more proofreading and address the typos in the revision.

---

> ### Author Response · Authors · 2021-08-17
> **Response to Reviewer 3bu4**
>
>
> Dear Reviewer 3bu4,
>
> We thank the reviewer time for the review, and we really hope to have a further discussion with reviewer 3bu4 to see if our response solves the concerns.
>
> We would sincerely appreciate it if reviewer 3bu4 could reply to the most important points in our rebuttal. For example, as we pointed out that for 20% and 10% training data schemes in Table 1, all reported numbers positively show clear performance improvements, which indicates sparse networks at both two sparsity levels benefit the data-limited GAN training.
>
> We genuinely hope reviewer 3bu4 could kindly check our response. Thank you!
>
> Best wishes,
>
> Authors

---

> > ### Comment · Reviewer_3bu4 · 2021-08-31
> > **Updated Score**
> >
> >
> > I was initially not satisfied with the method not improving the baseline on some cases, but after reading the added explanation I feel more convinced. Hence, I raised my score to 6.
> >
> > Best, Reviewer 3bu4

---

> > > ### Author Response · Authors · 2021-09-01
> > > **Response to Reviewer 3bu4**
> > >
> > > Dear Reviewer **3bu4**,
> > >
> > > Many thanks for all the helpful comments and positive re-assessment. We really appreciate reviewer **3bu4** for increasing our score.
> > >
> > > We are again very thankful for your time and support!
> > >
> > > Best wishes,
> > >
> > > Authors

---

### Author Response · Authors · 2021-08-23
**General Response**

Dear all reviewers:

We really appreciate all the reviewers for their valuable suggestions. We are thankful for the reviewers now appreciating this work’s novelty and practical usage.
- We thank reviewer **npsk** for strongly supporting the originality and significance of our paper.
- We thank reviewer **6Tg4** for the extremely helpful writing guidance and constructive experimental suggestions. We really appreciate reviewer **6Tg4** for increasing our score.
- We thank reviewer **Eqct** for the positive re-consideration after reading our rebuttals. We would appreciate it even more, if reviewer **Eqct** could take the valuable time to adjust his/her scores, that can more accurately reflect your current positive valuation of our work.
- We sincerely hope to have further discussion with reviewer **3bu4** to see if our response solves his/her concerns. We are confident that our response should have cleared the air, and we can clarify more if there is more need. We are happy to answer any additional questions and provide more information.

---

### Author Response · Authors · 2021-08-31
**Summary of our rebuttal**

We sincerely appreciate all reviewers’ and ACs’ time and efforts in reviewing our paper. We truly thank all for the insightful and constructive suggestions, which helped further improve our paper. We genuinely appreciate the positive **8-7-7-6** evaluation from reviewers **npsk**, **6Tg4**, **Eqct**, and **3bu4**.

Here is a summary of our updates:
- **[Additional Experiments]** As suggested by **AC**, reviewers **npsk**, **6Tg4** and **Eqct**, we conduct extra experiments on diverse sparsity levels, other pruning methods, new pretraining-and-transfer setting, and more baseline methods. All additional results consistently validate the effectiveness of our proposal.
- **[Writing]** We owe many thanks for reviewer **6Tg4**’s extremely helpful writing suggestions. All improved manuscript parts, together with other constructive discussions with reviewers **6Tg4** ,**Eqct**, and **3bu4**, will be delivered in our final version.

We really thank all reviewers’ and ACs' time and efforts again.

Best wishes,

Authors

---

### Decision · Program_Chairs · 2021-09-27

**Decision:**

Accept (Poster)

**Comment:**

This paper adopts the idea of pruning and data augmentation to improve the data efficiency of GANs. The rebuttal solves the reviewer's initial concerns about novelty and overfitting, and this paper is recommended for acceptance.